# Characterization of Metal-Specific T-Cells in Inflamed Oral Mucosa in a Novel Murine Model of Chromium-Induced Allergic Contact Dermatitis

**DOI:** 10.3390/ijms24032807

**Published:** 2023-02-01

**Authors:** Takamasa Yoshizawa, Kenichi Kumagai, Ryota Matsubara, Keisuke Nasu, Kazutaka Kitaura, Motoaki Suzuki, Yoshiki Hamada, Ryuji Suzuki

**Affiliations:** 1Department of Oral and Maxillofacial Surgery, School of Dental Medicine, Tsurumi University, Yokohama 230-8501, Japan; 2Department of Rheumatology and Clinical Immunology, Clinical Research Center for Rheumatology and Allergy, Sagamihara National Hospital, National Hospital Organization, Sagamihara 252-0392, Japan; 3Department of Oral and Maxillofacial Surgery, Dentistry and Orthodontics, The University of Tokyo Hospital, Tokyo 113-8655, Japan; 4Department of Oral and Maxillofacial Surgery, Sendai Tokushukai Hospital, Sendai 981-3116, Japan; 5Repertoire Genesis Inc., Osaka 567-0085, Japan; 6Department of Anatomy and Physiology, Faculty of Medicine, Saga University, Saga 849-8501, Japan

**Keywords:** chromium allergy, contact dermatitis, allergic contact mucositis, metal allergy

## Abstract

The element chromium (Cr) is a component of several types of alloys found in the environment, or utilized in dentistry, that may cause intraoral metal contact allergy. However, the pathological mechanism of intraoral Cr allergy remains unclear because there is no established animal model of Cr allergy in the oral mucosa. In this study, we established a novel murine model of Cr-induced intraoral metal contact allergy and elucidated the immune response in terms of cytokine profiles and T-cell receptor repertoire. Two sensitizations with Cr plus lipopolysaccharide solution into the postauricular skin were followed by a single Cr challenge of the oral mucosa to generate the intraoral metal contact allergy model. Histological examination revealed that CD3+ T-cells had infiltrated the allergic oral mucosa one day after exposure to the allergen. The increase in T-cell markers and cytokines in allergic oral mucosa was also confirmed via quantitative PCR analysis. We detected Cr-specific T-cells bearing TRAV12D-1-TRAJ22 and natural killer (NK) T-cells in the oral mucosa and lymph nodes. Our model demonstrated that Cr-specific T-cells and potent NKT-cell activation may be involved in the immune responses of Cr-induced intraoral metal contact allergy.

## 1. Introduction

Metal allergy is categorized as a delayed-type hypersensitivity (DTH) reaction triggered by antigenic protein with haptens that exert antigenicity. It can be caused by metal ions released by jewelry, footwear, preservatives, and cosmetics [1]. In addition to nickel (Ni), cobalt (Co), palladium (Pd), zinc (Zn), and chromium (Cr) have been reported to cause allergic contact dermatitis [2,3,4,5]. Cr hypersensitivity is one of the most prevalent occupational metal skin diseases in cement workers [6]. Occupational allergic contact dermatitis was the most frequently occurring allergy in construction workers (45%), and the most frequent allergen was chromium (Cr) in cement [6,7]. In addition, the production of a chronic generalized eczematoid reaction has been reported as a causal intraoral Cr-allergic contact dermatitis [8]. Recently, Cr has been widely used in dental restorations such as implants, crown prostheses, and dentures. However, the pathological mechanism of intraoral Cr allergy remains unknown because there is no established animal model of Cr allergy in the oral mucosa. According to previous studies of murine models of DTH in the oral mucosa, various chemicals, such as oxazolone (4-ethoxymethylene-2-phenyloxazol-5-one) and 2,4-dinitro-1-fluorobenzene, induce allergic contact mucositis (ACM) with the local accumulation of antigen-presenting cells and T-cells [9].

Typically, metal allergy is associated with acquired immunity, which facilitates the migration of metal-specific T-cells to the site of allergic inflammation. T-cells recognize antigens on antigen-presenting cells through surface-expressed T-cell receptors (TCRs), heterodimers composed of an α- and β-chain (TRA and TRB) that determine the high specificity of T-cells [10]. In previous studies, the cells in the peripheral blood and skin of patients with metal allergy had limited TCR repertoires [11]. Several novel murine models of Ni, Pd, Cr, and titanium (Ti)-induced allergic contact dermatitis (ACD) have been generated using footpad skin, and these have aided in the characterization of antigen-specific immune responses in terms of TCR usage [12,13,14,15]. These models enabled us to identify the accumulation of metal-specific T-cells in inflamed skin and to demonstrate that the restricted usage of TCR genes in metal allergy is a result of the prolonged exposure of the host immune system to putative metal-associated antigens. The analysis of the TCR repertoire enables the identification of antigen-specific T-cells [16]. Recent advances in next-generation sequencing (NGS) have permitted the quantitative analyses of the TCR repertoire using large amounts of TCR sequencing data [17,18].

In the present study, we established a novel murine model of Cr-induced allergy in the oral mucosa to examine how the accumulation of T-cells at the site of allergic inflammation contributes to the development of Cr allergy in the oral mucosa and how TCR gene usage is regulated.

## 2. Results

### 2.1. Oral Mucosa Swelling in Cr-Induced Allergic Mice

At Day 1 post-challenge, maximal swelling in the buccal area of the oral mucosa occurred in all mice (Figure 1). At Day 7 post-challenge, the swelling of the oral mucosa was significantly greater in the ACM mice than in the control mice, but it was not significantly different in irritant contact mucositis (ICM) mice. From Days 1 to 12 post-challenge, oral mucosa swelling in the ACM mice was greater than in the control mice.

### 2.2. Histological and Immunohistochemical Analyses of F4/80 and CD3 in the Oral Mucosa of Cr-Induced Allergic Mice

To determine whether macrophages and T-cells infiltrated the inflamed oral mucosa, histological and immunohistochemical (IHC) analyses were performed on the oral mucosa of the control, ICM, and ACM mice at Days 1 and 7 post-challenge. Hematoxylin and eosin (HE) staining revealed dense infiltration of inflammatory cells in the basal epithelial layer and upper dermis, in addition to a swelling of the oral mucosa, in the ACM and ICM mice, but not in the control mice (Figure 2A–E). Inflammatory cells accumulated in the epithelium and upper dermis of the ACM mice (Figure 2D). The partial separation of epidermal keratinocytes produced spongiotic dermatitis (Figure 2D). In the ICM mice, the inflammatory response in the oral mucosa was reduced (Figure 2B,C). In contrast, inflammation of the oral mucosa persisted for Day 7 post-challenge in the ACM mice (Figure 2E). We performed the IHC staining of CD3 and F4/80 in the oral mucosa of the control, ICM, and ACM mice to determine whether T-cells and macrophages had infiltrated the inflamed oral mucosa of the ACM mice (Figure 2F–O). Significant CD3+ T-cell infiltration occurred into the basal epithelial layer and upper dermis of the ACM mice by Day 1 post-challenge compared with the control and ICM mice (Figure 2F,G,I). The ACM mice retained CD3+ T-cells in the basal epithelial layer and upper dermis Day 7 post-challenge (Figure 2J). There was little infiltration of CD3+ T-cells into the basal epithelial layer and upper dermis of the control and ICM mice (Figure 2F–H). At Day 1 post-challenge, F4/80+ macrophages were predominant in the layer and upper dermis of the ACM mice (Figure 2N), but not in the control and ICM mice (Figure 2K,L,M,O).

### 2.3. mRNA Expression of T-Cell Markers in the Oral Mucosa of Cr-Induced Allergic Mice

We performed quantitative polymerase chain reaction (qPCR) of CD3, CD4, and CD8 to verify the infiltration of T-cells into the inflamed oral mucosa and determine their relative mRNA expression. CD3 expression in the ACM mice was significantly higher than in the control and ICM mice at Days 1 and 7 post-challenge (Figure 3). At Day 1 post-challenge, the CD8/CD4 ratio was significantly higher in the ACM mice than in the control and ICM mice (Figure 3).

### 2.4. Relative mRNA Expression of T-Cell-Related Cytokines in the Oral Mucosa of Cr-Induced Allergic Mice

Subsequently, using qPCR analysis, the mRNA expression of inflammatory markers was analyzed to examine inflammation in the allergic oral mucosa. We compared the expression levels of a proinflammatory cytokine (IL-1β), a Th1-related gene (IFN-γ), a Th2-related gene (IL-4), and the serine protease(granzyme B) in the oral mucosa of the control, ICM, and ACM mice using qPCR (Figure 4). At Days 1 and 7 post-challenge, IL-1β and granzyme B expression levels were significantly higher in the ACM mice than in the control and ICM mice. IFN-γ, in contrast to IL-4, was significantly higher in the ACM mice relative to the ICM mice with similar expression maintained on Days 1 and 7.

### 2.5. TCR Repertoire Usage in the Oral Mucosa and Cervical Lymph Nodes of the ICM and ACM Mice

We performed an NGS-based TCR repertoire analysis to determine the diversity of T-cells that had infiltrated the oral mucosa and cervical lymph nodes of the ICM and ACM mice at Day 1 post-challenge. A 3D representation of the TRA repertoire revealed the dominance of particular combinations of TRAV and TRAJ genes, as well as the breadth of TCR usage diversity (Figure 5). The 3D images of the TRA repertoire revealed a low level of expression accompanied by a broad distribution of TRAV and TRAJ. The usage of TRAV11d-TRAJ18 was considerably higher in the oral mucosa and cervical lymph nodes of the ICM and ACM mice († in Figure 5).

Next, we analyzed the CDR3 amino acid sequences shared by the ICM and the ACM mice (Figure 6). Notably, the proportion of T-cells bearing TCR TRAV11d-TRAJ18 (CVVGDRGSALGRLHF) was highest in the oral mucosa and cervical lymph nodes (Figure 6, green shading) compared with other T-cells. In mice, invariant natural killer T (iNKT) cells express a TRA encoded by the gene segments TRAV11d-TRAJ18 [19]. T-cells bearing TRAV12D-1-TRAJ22 (Figure 6, yellow shading) with common CDR3 amino acid sequences (CALSEKSSGSWQLIF) were detected frequently in the oral mucosa and cervical lymph nodes of the ACM mice but in only a few cervical lymph nodes of the ICM mice. Furthermore, to identify changes in the T-cell diversity at Days 1, 3, and 7 post-challenge, we examined the TCR repertoire of T-cells in Cr allergy that had infiltrated the oral mucosa of the ACM mice at Days 3 and 7 post-challenge (Appendix A). The frequency ranking of TRA clonotypes for the top 30 read percentages indicated a high proportion of iNKT-cells in the oral mucosa of the ACM mice at Days 3 and 7 post-challenge (Appendix A, green shading). T-cells bearing TRAV1-TRAJ33 with common CDR3 amino acid sequences (CAVRDSNYQLIW) were detected in the oral mucosa of the ACM mice between Days 3 and 7 post-challenge (Appendix A, blue shading). Mucosal-associated invariant T (MAIT) cells in mice express a TRA encoded by the TRAV1-TRAJ33 gene segments [19]. Next, we examined the common CDR3 amino acid sequences in the TRB repertoire in the ACM mice (Appendix A). There was no shared TRB clone in the sequences from the inflamed oral mucosa of the ACM mice at Days 1, 3, and 7 post-challenge.

## 3. Discussion

In this study, we successfully established a mouse model of intraoral Cr allergy and induced a delayed allergic response in the oral mucosa, building on previous studies documenting a Cr-induced allergic mouse model using the footpad skin or the auricle [15,20]. Recently, the usage of Cr in dental treatments, such as implants, and other intraoral treatments has increased. In a previous study, adverse effects were observed when local chromium concentrations were high owing to the release of particulate and soluble chromium released from implants [21]. However, the pathological mechanism of intraoral Cr allergy remains unclear because there is no established animal model of Cr allergy in the oral mucosa.

To our knowledge, this is the first study to clarify Cr-specific immune responses in the oral mucosa using a murine model of intraoral Cr allergy. In the ICM group, which was exposed to the external stimuli of Cr injections without sensitization, histopathological examination revealed a significant amount of inflammatory cell infiltration under the epithelium, as well as inflammatory reactions in the dermis and muscle layer at Day 1 post-challenge. In the ICM group at Day 7 post-challenge, there was little irritated inflammation at the site. The ACM group showed obvious spongiform edema and T-cell infiltration at Day 1 post-challenge in the injected epithelium. Intraepithelial spongiform edema is a defining feature of DTH [22]. By Day 7 post-challenge, the spongiform edema had decreased and T-cells had infiltrated the subepithelial layer. In the ACM group, F4/80+ macrophages infiltrated the basal epithelial and subepithelial layers at Day 1 post-challenge but then decreased until Day 7 post-challenge. Our Cr-induced allergic mouse model in the oral mucosa has characteristics resembling DTH; therefore, it may be appropriate for studying delayed allergic reactions in the oral mucosa. As regards the duration of acute ACD, previous studies have indicated that it develops between 24 and 48 h after initiation [23]. Initially, skin lesions are asymmetric and limited to the area of contact; later, they often spread or disseminate. In the case of severe reactions, swelling and blistering are observed. The major clinical differences between non-sensitized irritant contact dermatitis (ICD) and sensitized ACD are the more rapid onset of ICD and the tendency of ACD to spread. Characteristic widespread reactions are usually symmetric, although the primary reaction is not [23]. Metal allergy can be induced by either CD4+ or CD8+ T-cells, depending on the antigen processing pathway [9,24]. In this mouse model of intraoral Cr allergy, CD8+ T-cells accumulated in large numbers in ACM at Day 1 post-challenge, but this was not observed in control or ICM mice (Figure 3). These results suggest that sensitization had occurred and that CD8+ T-cells specific to Cr were induced in intraoral Cr-induced allergic mice. Another mouse model of metal allergy found that CD8+ T-cells accumulated at inflammatory sites, suggesting that these T-cells promote inflammation during the DTH induction stage [14].

We also compared the IL-1β, IFN-γ, IL-4, and granzyme B expression levels in the oral mucosa of the control, ICM, and ACM mice (Figure 4). At Day 1 post-challenge, the levels of IFN-γ expression in the ACM mice were significantly higher than those seen in the ICM mice. However, there were no differences in the levels of IL-4 expression between the ACM and ICM mice at Day 1 post-challenge. This indicates that the Cr-allergic immune response in the oral mucosa may be Th-1-biased at Day 1 post-challenge. CD8+ T-cells were reported as infiltrating the inflammatory site during metal allergy induction and producing IFN-γ [14]. Furthermore, IL-1β induces Th-1 and Th-17 differentiation and stimulates cytokine production in T-cells and CD8+ T-cells residing in the epidermal tissue, which may increase IL-1β production via a positive feedback loop [25]. Our results were consistent with those of this study. According to a separate study, granzyme B expression was elevated in the skin of patients with metal allergy [26]. CD8+ T-cells are cytotoxic and kill target T-cells by releasing cytotoxic granules. Our findings indicate that IFN-γ is the principal effector cytokine of Cr allergy in the oral mucosa. The apoptosis of keratinocytes induced by macrophages and CD8+ T-cells may also play a role in the pathogenesis of Cr allergy in the oral mucosa. We previously developed a mouse model of Cr-induced ACD in the footpad of mice and found that allergic-specific T-cells used a specific TCR repertoire in Cr-induced ACD [15]. The infiltrating T-cells included iNKT-cells and Cr-specific T-cells with VA11-1/VB14-1 usage. In this study, we observed an accumulation of iNKT-cells in the inflamed mucosa of the ICM and ACM mice. These iNKT-cells help to amplify early immune responses [19,27]. The accumulation of iNKT-cells in the inflamed oral mucosa in Ni allergy suggests that iNKT-cells are involved in metal allergy [28]. In this study, we also observed the accumulation of Cr-specific T-cells bearing TRAV12D-1-TRAJ22 and iNKT-cells in the inflamed oral mucosa of Cr-allergic mice (Figure 5 and Figure 6). This result contributes significantly to our understanding of antigen recognition by Cr-specific T-cells. Th1-type cytokines and cytotoxic molecules from iNKT-cells and Cr-specific T-cells are likely to be positively correlated with the pathogenesis of metal allergy in the footpads of mice. TCR repertoire and CDR3 sequencing analyses revealed a shared TCR repertoire expressing TRAV12D-1-TRAJ22 with the CDR3 sequence (CALSEKSSGSWQLIF) in the oral mucosa and lymph nodes at Day 1 post-challenge (Figure 6). In contrast to the highly restricted TRAV repertoire, oral mucosa-infiltrating T-cells displayed a relatively broad TRBV repertoire. As the TCR repertoire involved in Cr allergy differs between the oral mucosa, the footpad, and the ears, it is likely that organ-specific immune responses may occur during metal allergy. Furthermore, iNKT-cells were abundant in the ICM and ACM mice, while MAIT-cells were detected in the oral mucosa of the ACM mice at Days 3 and 7 post-challenge, whereas TRAV12D-1-TRAJ22-bearing T-cells were not (Appendix A). iNKT and MAIT-cells were found in high numbers in our previous research on cross-reactive metal allergy, suggesting that they may be involved in the development of cross-reactive metal allergy [29]. Consequently, the data on the TCR repertoire contribute to our understanding of the structural identity of antigenic determinants recognized by Cr-induced intraoral metal contact allergy.

In conclusion, we demonstrated that CD8+ T-cells infiltrated the oral mucosa during an allergic inflammatory response in a mouse model of metal allergy. We also identified clones of T-cells specific to Cr-induced allergy. Our analysis of the TCR repertoire demonstrated that restricted TRAV12D-1-TRAJ22 usage with the CDR3 amino acid sequence (CALSEKSSGSWQLIF) and broad TRBV might specifically recognize Ag in Cr allergy of the oral mucosa. In that case, the direct cloning of TCR genes from local sites of inflammation using this model would be a powerful tool for advancing our understanding of T-cell-mediated immune disease in metal allergy, and it would also provide new insights into Ag recognition by Cr-specific TCR in the oral mucosa.

## 4. Materials and Methods

### 4.1. Animals

*BALB/cAJcl* mice (4-week-old females) were purchased from CLEA Japan (Tokyo, Japan). All of the mice were in good health throughout the duration of the study and were given 1 week to acclimatize to their surroundings before the study began. At the beginning of the experiment, the mice were 5 weeks old. All of the mice were kept in plastic cages (with lids made of stainless-steel wire) and given ad libitum access to food and water. The mice were kept in our conventional animal facility with a temperature of 19–23 °C, a humidity of 30–70%, and a 12 h day/night cycle. During experiments, their appearance, their behavior, and the amount of food and water they consumed were monitored daily to ensure the health and comfort of all of the mice. All surgeries were performed using three different types of mixed anesthetics, and every effort was made to minimize animal suffering. All of the mice were euthanized by cervical dislocation under three different types of mixed anesthetic agents to prevent pain caused by tissue harvesting.

### 4.2. Reagents

FUJIFILM Wako Pure Chemical Co., Ltd. was the supplier of high-purity CrCl_2_ (>95%) (Osaka, Japan). Lipopolysaccharide (LPS) from *Escherichia coli* (O55:B5) prepared by phenol–water extraction was purchased from Sigma-Aldrich (St Louis, MO, USA). CrCl_2_ and LPS were dissolved in sterile saline (Otsuka Normal Saline, Otsuka Pharmaceutical Factory, Inc., Tokushima, Japan).

### 4.3. Anesthetic Agents

The anesthetic was created by combining three medications. We purchased medetomidine hydrochloride from Nippon Zenyaku Kogyo Co., Ltd. (Fukushima, Japan), midazolam from Sandoz (Tokyo, Japan), and butorphanol tartrate from Meiji Seika Pharma Co., Ltd. (Tokyo, Japan). These drugs were kept at room temperature (RT). We combined doses of 0.3 mg/kg medetomidine hydrochloride, 4 mg/kg midazolam, and 5 mg/kg butorphanol tartrate. The concentration ratio of the three types of mixed anesthetic agents was determined according to data from a previous study [30]. Typically, 25 mL of anesthetic agent was prepared by combining 0.75 mL of medetomidine hydrochloride, 2 mL of midazolam, 2.50 mL of butorphanol tartrate, and 19.75 mL of sterile saline. All of the agents were diluted in sterile saline and stored in the dark at 4℃. The mice were administered a volume of 10 µL/g of body weight of the anesthetic mixture. The mixture of the three types of anesthetic agents was injected intraperitoneally into every mouse.

### 4.4. Experimental Protocol for the Mouse Model of Cr-Induced Intraoral Metal Contact Allergy

We have developed an experimental protocol for the induction of metal allergy in the oral mucosa [28] based on a previous protocol for the induction of metal allergy in footpad skin. Mice were separated into three groups: ACM (n = 15), ICM (n = 15), and control (n = 15), with each group consisting of randomly selected mice. All experiments were performed in another room upon transfer from the animal holding area.

Sensitization: The mice received two intradermal injections of 125 µL of 10 mM CrCl_2_ and 10 g/mL LPS in sterile saline with a 7-day interval between injections. The ACM group was sensitized on the postauricular skin in the left and right ears. Seven days after the second sensitization, the mice were challenged for the first time.

Challenge for elicitation: 25 µL of 10 mM CrCl_2_ without LPS in sterile saline was used to elicit an immune response. The ACM mice were challenged by submucosal injection in the left and right buccal regions of the oral mucosa. The ICM mice were not sensitized, but they were challenged with CrCl_2_. The control mice were sensitized and then challenged with sterile saline.

### 4.5. Measurement of Oral Mucosa Swelling

The swelling of the buccal region was measured before the challenge and at 1, 2, 3, 5, 7, 10, 12, and 14 days after the initial challenge using a Peacock Dial Thickness Gauge (Ozaki MFG Co., Ltd., Tokyo, Japan). The thickness of the buccal area oral mucosa was measured before and after the challenge, and the difference was recorded. The same experimenter performed all procedures on the mice while they were under anesthesia.

### 4.6. Histological and IHC Analysis

For histological and immunohistochemical analyses, buccal oral mucosa samples were obtained from the control, ICM, and ACM mice. Furthermore, the tissue samples were immersed in 4% paraformaldehyde–lysine–periodate for 48 h at 4 °C. After a 10-min wash in PBS, fixed tissues were soaked in 5% sucrose/PBS for 1 h at 4 °C, 15% sucrose/PBS for 3 h at 4 °C, and finally, 30% sucrose/PBS overnight at 4 °C. The buccal mucosa tissues were snap-frozen by immersion in a mixture of acetone and dry ice in Tissue Mount (Chiba Medical, Saitama, Japan). Frozen sections were cut into 6 µm thick cryosections and air-dried on poly-L-lysine-coated glass slides. The HE stain was applied to cryosections for histological analysis. The cryosections were stained with anti-mouse F4/80 (1:1000; Cl-A3-1, Abcam, Cambridge, UK) and anti-mouse CD3 (1:500; SP7, Abcam) monoclonal antibodies for IHC analysis (mAbs). The F4/80 monoclonal antibody was used to detect mouse macrophage populations in many buccal oral mucosal tissues. The sections were incubated at RT for 30 min in PBS containing 5% normal goat/rabbit serum, 0.025% Triton X-100 (FUJIFILM Wako Pure Chemical, Osaka, Japan), and 5% bovine serum albumin (Sigma-Aldrich). Sections were incubated with primary mAbs for 1 h at RT. After three 5-min washes with PBS, intrinsic peroxidase was inhibited with 3% H_2_O_2_ in methanol. After the tissue sections were soaked in distilled water, they were washed twice and incubated for 1 h at RT with a secondary antibody (biotinylated goat anti-hamster immunoglobulin G or biotinylated rabbit anti-rat immunoglobulin G). The sections were treated with Vectastain ABC Reagent (Vector Laboratories, Burlingame, CA, USA) for 30 min at RT, followed by 3,3-diaminobenzidine staining (0.06% diaminobenzidine and 0.03% H_2_O_2_ in 0.1 M Tris–HCl, pH 7.6; FUJIFILM Wako Pure Chemical). Hematoxylin was used to counterstain tissue sections to visualize cell nuclei.

### 4.7. RNA Extraction

The buccal region of the oral mucosa in each mouse was freshly obtained and immersed immediately in RNAlater RNA Stabilization Reagent (Invitrogen, Carlsbad, CA, USA). Total RNA was extracted from the buccal region of the oral mucosa using the RNeasy Lipid Tissue Mini Kit (Qiagen, Hilden, Germany) as directed by the manufacturer. Complementary DNA (cDNA) was synthesized from DNA-free RNA.

### 4.8. Quantitative Polymerase Chain Reaction

The expression levels of immune response-related genes, including T-cell-related CD antigens, cytokines, cytotoxic granules, and regulatory T-cell transcription factors, were measured using qPCR on a Bio-Rad CFX96 instrument (Bio-Rad, Hercules, CA, USA). Previously, specific primers for GAPDH, CD3, CD4, CD8, IFN-γ, Granzyme B, IL-4, and IL-1β were described [15,16,31]. The Prime Script RT reagent Kit (Takara Bio, Shiga, Japan) was used to convert newly isolated total RNA from the oral mucosa and submandibular lymph nodes into complementary DNA (cDNA). In a final volume of 10 µL, the PCR contained 5 µL SsoFast EvaGreen Supermix (Bio-Rad), 3.5 µL RNase/DNase-free water, 0.5 µL of 5 µM primer mix, and 1 µL cDNA. The following cycling conditions were used: 30 s at 95 °C, followed by 50 cycles of 1 s at 95 °C and 5 s at 60 °C. At the conclusion of each protocol, a melting curve analysis from 70 °C to 90 °C was performed to confirm the homogeneity of the PCR products. All tests were performed three times, and the mean values were used to determine gene expression levels. Five 10-fold serial dilutions of each standard transcript were utilized to determine the absolute amount, specificity, and amplification efficiency of each primer set. Standard transcripts were generated through the in vitro transcription of the corresponding PCR product in a plasmid. DNA sequencing confirmed the nucleotide sequences using the CEQ8000 Genetic Analysis System (Beckman Coulter, Fullerton, CA, USA). Next, using an Agilent DNA 7500 Kit on an Agilent 2100 Bioanalyzer, their quality and concentration were determined (Agilent, Santa Clara, CA, USA). The expression of the GAPDH gene served as an internal control.

### 4.9. Mouse TCR Repertoire Analysis

Total RNA was extracted from the oral mucosa and cervical lymph nodes of the ICM and ACM mice at Day 1 post-challenge and from the oral mucosa of the ACM mice at Days 3 and 7 post-challenge. NGS was used to perform TCR repertoire analysis developed by Repertoire Genesis Inc. (Osaka, Japan [18]). As detailed in a previous report, an unbiased adaptor-ligation PCR was performed [32]. Superscript III reverse transcriptase was utilized to convert total RNA to cDNA (Invitrogen). Subsequently, double-stranded (ds) cDNA was synthesized, and an adaptor was ligated to the 5′ end of the ds-cDNA before it was cut with the SphI restriction enzyme. For TCRα, PCR was performed using a P20EA adaptor primer and a TCR α-chain constant region-specific primer (mCA1). The second PCR was conducted using the same PCR conditions and primers, mCA2 and P20EA. As regards TCRβ, the first and second primers used for PCR were mCB1 and mCB2, respectively. After Tag PCR amplification, index (barcode) sequences were amplified using a Nextera XT Index Kit v2 setA (Illumina, San Diego, CA, USA). Sequencing was performed using the paired-end Illumina MiSeq platform (2 × 300 base pairs [bp]). The repertoire analysis software of Repertoire Genesis, Inc. was used for automatic data processing, assignment, and aggregation. TCR (TRA and TRB) sequences were mapped to a reference sequence dataset from the international ImMunoGeneTics information system (IMGT) database (http://www.imgt.org, accessed on 1 November 2022 [33]). Nucleotide sequences of CDR3 regions ranged from a conserved cysteine at position 104 (Cys104) to a conserved phenylalanine at position 118 (Phe118), and the following glycine (Gly119) was translated into an amino acid sequence. A unique sequence read was defined as a sequence read with no identity in TRAV, TRAJ, and the deduced amino acid sequence of CDR3. The copy number of identical unique sequence reads in each sample was automatically counted and ranked by copy number using software for repertoire analysis. The percentage occurrence frequencies were calculated for sequence reads containing TRAV and TRAJ and for genes.

### 4.10. Statistical Analysis

Statistically significant differences between the mean values of each experimental group were analyzed using the Kruskal–Wallis test followed by Dunn’s multiple comparison tests. All analyses were performed using IBM SPSS Statistics version 24 (IBM, Armonk, NY, USA). A *p*-value of <0.05 was considered significant.

## Figures and Tables

**Figure 1 ijms-24-02807-f001:**
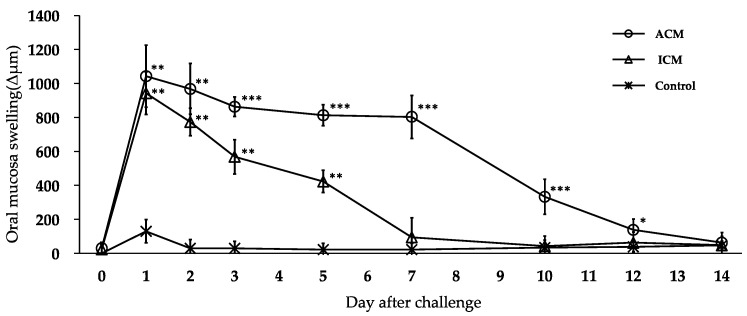
Swelling of the oral mucosa in Cr-induced allergic mice. The oral mucosa of all mice reached maximal swelling at Day 1 post-challenge. The oral mucosa swelled significantly more in the ACM mice from Days 1 to 12 post-challenge compared with the control mice. The bars and error bars represent the mean plus the standard deviation. The Kruskal–Wallis test was used to determine statistical significance, followed by Dunn’s multiple comparison tests (* *p* < 0.05, ** *p* < 0.01, *** *p* < 0.001).

**Figure 2 ijms-24-02807-f002:**
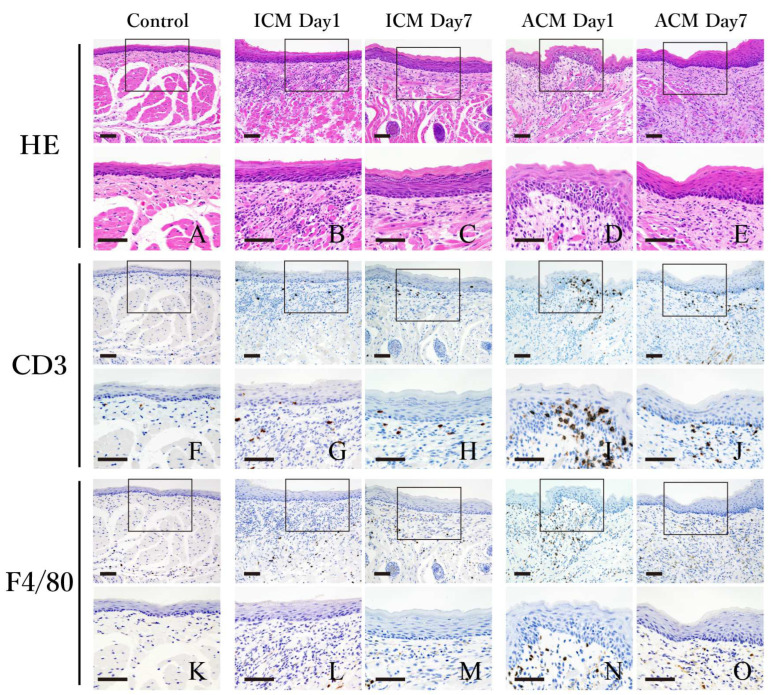
Histological and IHC analyses of CD3 and F4/80 in the oral mucosa of the control, ICM, and ACM mice. Histological and IHC analyses of CD3+ and F4/80+ T-cells in the oral mucosa at Days 1 and 7 post-challenge. At Day 1 post-challenge, the ACM mice exhibited abundant infiltration of mononuclear cells, swelling of the mucosal epithelium, and epidermal spongiosis (**D**), and IHC analyses revealed the presence of CD3+ T-cells in the epithelium of the ACM mice (**I**). Sections were stained with HE (**A**–**E**), CD3 (**F**–**J**), and F4/80 (**K**–**O**). Scale bar = 10 µm.

**Figure 3 ijms-24-02807-f003:**
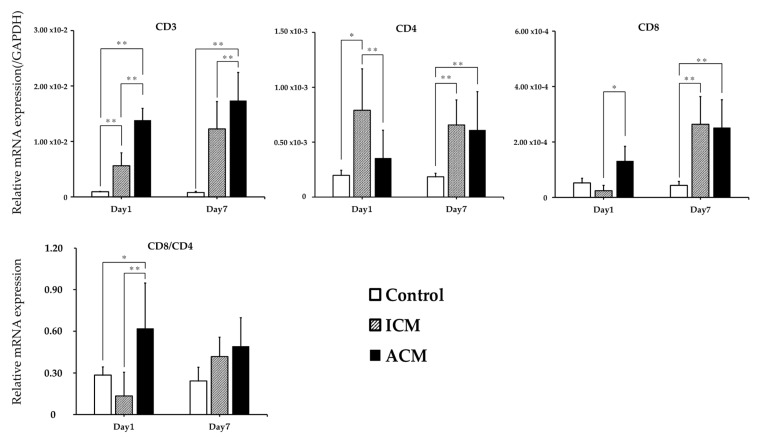
mRNA expression of T-cell phenotypes in the oral mucosa of Cr-induced allergic mice. At Days 1 and 7 post-challenge, the mRNA expression of CD3, CD4, CD8, and the CD8/CD4 ratio in the oral mucosa was evaluated (n = 8). The expression of the GAPDH gene served as an internal control. The mean and standard deviation are represented by bars and error bars. Kruskal–Wallis and Dunn’s multiple comparison tests were used to determine statistical significance (* *p* < 0.05, ** *p* < 0.01).

**Figure 4 ijms-24-02807-f004:**
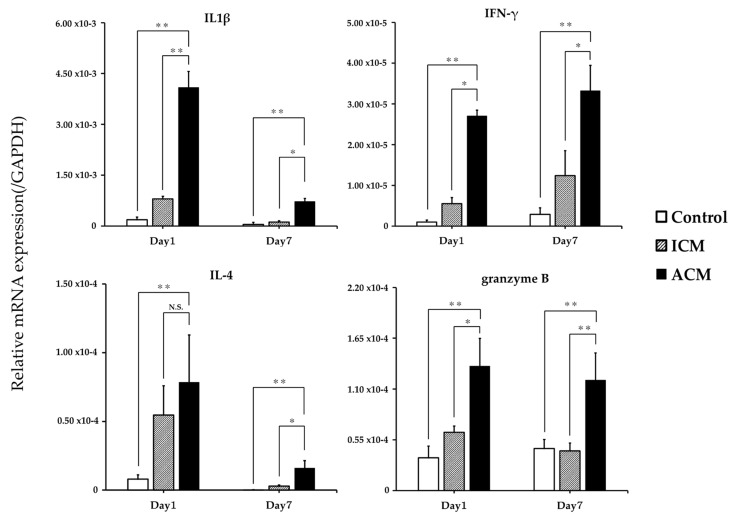
mRNA expression of T-cell-related cytokines in control, ICM, and ACM mice. At Days 1 and 7 post-challenge, the mRNA expression levels of IL-1β, IFN-γ, IL-4, and granzyme B in the oral mucosa (n = 8) were evaluated. The expression of the GAPDH gene served as an internal control. The mean and standard deviation are represented by bars and error bars, respectively. Kruskal–Wallis and Dunn’s multiple comparison tests were used to determine the statistical significance of differences (* *p* < 0.05, ** *p* < 0.01, N.S. = not significant).

**Figure 5 ijms-24-02807-f005:**
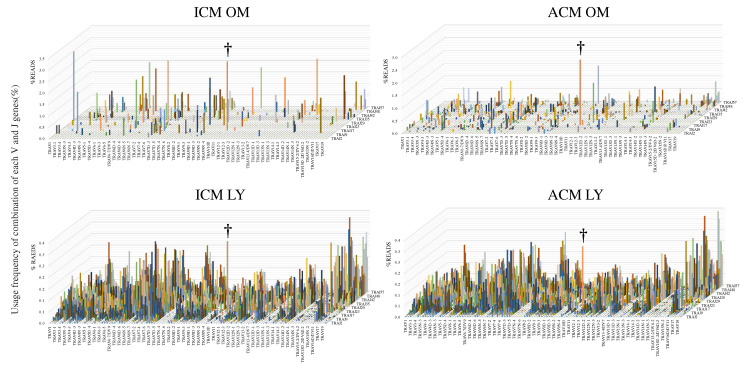
TCR repertoire in the oral mucosa and cervical lymph nodes of the ICM and ACM mice. NGS-based TCR repertoire analysis was performed on the oral mucosa and cervical lymph nodes of the ICM and ACM mice at Day 1 post-challenge (n = 6). Additionally, at Day 1 post-challenge, 3D images of the TCR repertoire depict the skewing of T-cells infiltrating the oral mucosa and cervical lymph nodes of the ICM and ACM mice. Combining TRAV on the X-axis and TRAJ on the *Z*-axis, with the frequency (percentage) of each clone on the Y-axis, the 3D images depict the TCR repertoire. †: TRAs bearing TRAV11d-TRAJ18. OM: oral mucosa. LY: lymph nodes of the cervical region.

**Figure 6 ijms-24-02807-f006:**
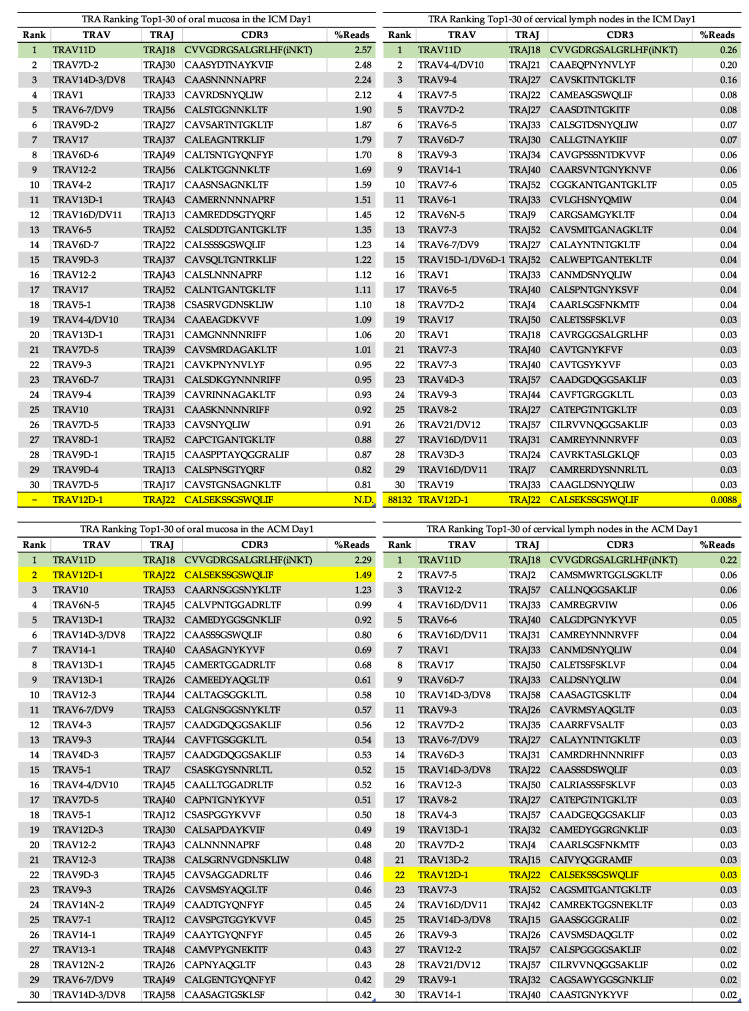
Ranking of the top 30 most frequently read TRA clonotypes (as a percentage) in the oral mucosa and cervical lymph nodes of the ICM and ACM mice at Day 1 post-challenge. Distributions of frequency (%) reads of amino acid sequences of CDR3 regions in the oral mucosa and cervical lymph nodes of the ACM mice revealed the presence of Cr-specific T-cells bearing TRAV12D-1-TRAJ22 (yellow shading). In both the oral mucosa and the cervical lymph nodes of the ICM and ACM mice, iNKT-cells (green shading) are highly expressed. N.D. = not detected.

## Data Availability

The data presented in this study are available on request from the corresponding author.

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
