# Peer review of "Characterization of Metal-Specific T-Cells in Inflamed Oral Mucosa in a Novel Murine Model of Chromium-Induced Allergic Contact Dermatitis"

_ijms, 2023, doi:10.3390/ijms24032807_

Round 1
Reviewer 1 Report
Please open the attached file.

Reviewer 2 Report
In this paper the authors present a novel murine model of chromium-induced allergic contact dermatitis. The manuscript is well written and structured. I have several minor suggestions.
In the Introduction Section, the authors should include more data on intraoral metal allergy, focusing on the one induced by chromium. An interesting article is doi: 10.3390/dj8030083
In the Discussion Section, the authors should discuss more about the toxic effects of chromium. It would be useful to include data from this article, doi: 10.5772/intechopen.90347
The authors mention the terms "a delayed-type hypersensitivity" and "type IV allergy". They should specify that type IV hypersensitivity reactions are also known as delayed-type hypersensitivity reactions.
Lines 267-270 "The stimulated inflammation at the inflammation site gradually decreased in the ICM group 7 days post challenge". Please rephrase it.
Round 2
Reviewer 1 Report
The authors satisfactorily addressed all comments and suggestions. Thank you.